# Experiences of trial participants and site staff of participating in and running a large randomised trial within fertility (the endometrial scratch trial): a qualitative interview study

Robin Chatters ,[1] David White,[1] Clare Pye,[2] Ana Petrovic,[3] Anya Sizer,[4] Pavithra Kumar,[1] Mostafa Metwally[2]

¹Sheffield Clinical Trials Research Unit, School of Health and Related Research, The University of Sheffield, Sheffield, UK
²Obstetrics, Gynaecology & Neonatology, Sheffield Teaching Hospitals NHS Foundation Trust, Sheffield, UK
³Rotherham Doncaster and South Humber Mental Health NHS Foundation Trust, Doncaster, UK
⁴Fertility Network UK, London, UK

**Correspondence to**
Mr Robin Chatters;
r.chatters@sheffield.ac.uk

## ABSTRACT

**Objectives** To explore the experiences of endometrial scratch (ES) trial participants and site staff of trial recruitment and participation, in order to improve the experience of participants in future trials.

**Design** Qualitative study of a subset of participants in the ES randomised controlled trial and a subset of trial site staff.

**Setting** A purposeful sample of 9 of the 16 UK Fertility Units that participated in the trial.

**Participants** A purposeful sample of 27 trial participants and 7 site staff.

**Results** Participants were largely happy with the recruitment practices, however, some were overwhelmed with the amount of information received. Interviewees had positive preconceptions regarding the possible effect of the ES on the outcome of their in vitro fertilisation (IVF) cycle, which often originated from their own internet research and seemed to be exacerbated by how site staff described the intervention. Some participants appeared to not understand that receiving the ES could potentially reduce their chances of a successful IVF outcome. Those randomised to the control arm discussed feeling discontent; site staff developed mechanisms of dealing with this.

**Conclusions** A lack of equipoise in both study participants and the recruiting site staff led to trial participants having positive preconceptions of the potential impact of the ES on their upcoming IVF cycle. Trial participants may not have understood the potential harms of participating in a randomised trial. The trial information sheet did not clearly state this; further research should assess how such information should be presented to potential participants, to proportionately present the level of risk, but to not unduly discourage participation. The amount of information fertility patients require about a research study should also be investigated, in order to avoid participants feeling overwhelmed by the amount of information they receive prior to starting IVF.

**Trial registration number** ISRCTN23800982.

## INTRODUCTION

A wealth of literature exists that qualitatively assesses the experiences of participants being

### STRENGTHS AND LIMITATIONS OF THIS STUDY

⇒ We present data from a purposeful sample of 27 trial participants and 7 site staff regarding their experiences of taking part in, and running, the endometrial scratch randomised controlled trial.
⇒ As far as we are aware, this is the first study to investigate the experience of fertility patients of being recruited to, and participating in, a randomised trial.
⇒ We included both participants randomised to the control and intervention arms of the trial, plus those who did and did not have a live birth after their in vitro fertilisation cycle.
⇒ Recall may be an issue, as trial participants were interviewed on average 17.8 months after they were recruited to the trial.
⇒ We only interviewed individuals who participated in the trial, and therefore, interviews may represent a biased view of the experiences of being recruited to and participating in the trial.

recruited to trials across various disease areas, indicating that participants can find it difficult to understand the potential harms of participating in a study,[1 2] and staff experience challenges in expressing equipoise to participants.[3] The participant information sheet (PIS) has been extensively researched; in one study the 'traditional' PIS was found to oversupply information to 79.5% of participants.[4] Other studies have identified that, if given a choice, participants often opt for a shorter version of the PIS.[5 6] To our knowledge, the experience of patients participating in randomised trials in fertility has not been explored, although recruitment techniques have been described across two trials.[7]

The endometrial scratch (ES) trial was a randomised controlled trial (RCT) that aimed to assess the clinical and cost effectiveness of undertaking the ES prior to first

time in vitro fertilisation (IVF). Participants (women recruited between July 2016 and October 2018 from 16 fertility centres across the UK) were randomly allocated to either receive usual IVF treatment, or the ES procedure, followed by usual IVF treatment. The ES involves scratching the lining of the womb (the endometrium) prior to a cycle of IVF and is hypothesised to improve the chances of the embryo implanting by improving the receptivity of the endometrium.[8] A lack of good-quality evidence exists to support the delivery of the ES in women undergoing their first cycle of IVF.[9] The full trial protocol is published elsewhere.[10]

There are several reasons to investigate the experience of participants and staff within this trial. Undergoing IVF can be a distressing and emotional experience that involves receiving invasive procedures, causing physical and mental stress.[11 12] Patients receive a lot of information prior to their treatment, which has been associated with anxiety.[13] It is, therefore, important to ensure that participation in research does not create an extra burden for the patient. Furthermore, as women undergoing fertility treatments often seek information from sources other than their fertility team (eg, the internet, books and magazines), it is pertinent to ascertain how this affected the participant's experience of recruitment to the trial.[14] The study involved potentially difficult decisions for participants, specifically, whether to participate in the trial given the 50% chance of being randomised to a novel intervention with unknown benefits or harms, or the treatment as usual control group. Qualitative evaluations in other trial populations have identified that participants often feel discontent when randomised to the treatment as usual arm.[15] Given that recruiting staff have been found to struggle to convey equipoise in previous studies,[3 16] it is interesting to evaluate if these individuals conveyed equipoise, and if this impacted on the participant's experiences of the trial.

The aim of this qualitative study was to understand the experiences of participants being recruited to—and participating in—the ES trial, including any issues or barriers they faced. We also aimed to understand trial site staff's experiences, with a general aim to improve the experience for participants in future trials.

## METHODS

### Sampling, approach and recruitment

Purposive sampling was used to achieve the maximum variation in characteristics across trial participants and staff. First, six trial sites were selected, by seeking distribution of the characteristics described in table 1 (site sampling attributes). Trial participants who had recently completed the trial and had not withdrawn from trial follow-up were then sampled from these sites, using the attributes presented in table 1 (participant sampling attributes).

Site staff (principal investigators (PIs) and research nurses (RNs)) from the six selected centres were initially sampled for the staff interviews. Due to low staff availability and willingness to participate in the interviews, additional sites were selected as long as the research nurse was still in post and had undertaken recruitment to the trial.

Potential interviewees were initially approached by letter or email. If no contact was received, individuals were telephoned or emailed approximately 1 week after the initial contact was made. Verbal consent was obtained immediately prior to the interview via telephone, with the conversation regarding consent being audiorecorded.

### Research team and reflexivity

One woman (AP, research assistant with a PhD) and one man (RC, trial manager with a BSc) conducted the interviews with all participants. Both had prior training and experience of undertaking qualitative interviews. One other female researcher (PK, research assistant with an MSc) assisted with data analysis, and a female patient and public involvement (PPI) representative (AS, with experience of fertility treatments and an employee of a patient charity—Fertility Network UK) reviewed the interview schedule and this manuscript.

RC managed the ES Trial from the start; AP commenced her involvement more recently prior to the start of this

| Table 1 Site and participant level sampling attributes | |
|---|---|
| **Attribute** | **Categories** |
| Site sampling attributes | |
| Site type | NHS site/privately owned |
| Size of centre | Number of IVF cycles undertaken per year |
| Consent rate of centre | The proportion consented relative to those screened |
| Participant sampling attributes | |
| Intervention status | ES received, ES declined, treatment as usual |
| Age | 30 or below, above 30 |
| Duration of infertility | Less than 36 months, 36 months or more |
| Outcome of first cycle of IVF | Live birth, no live birth |
| ES, endometrial scratch; IVF, in vitro fertilisation; NHS, National Health Service. | |

qualitative study. A relationship between site staff and RC was therefore already formed, which was also true for AP but to a lesser extent. Site staff would have been aware of the interviewer's involvement in the design and delivery of the trial and their reasons for doing the research.

The interviewers had no involvement in the clinical care of the trial participants, nor had they had previous contact. Participants would have been aware of the interviewer's reasons for undertaking the research and may or may not have been aware of the interviewers' involvement in the trial in which they participated.

### Data collection
A semistructured interview schedule was used to guide the interviews—with separate schedules being used for trial participants and site staff (see online supplemental files 1; 2). The schedules were pilot tested in one interview per schedule, and adjustments were made following discussions between AP and RC; the pilot interview is included in the analysis. The participant interviews were undertaken prior to the staff interviews in order to allow themes from the former to be discussed with staff. Prompts were used during interviews and repeat interviews were not carried out.

Interviews were audiorecorded and transcribed verbatim. There were no other individuals present at the interviews, except for, in a few instances, the participant's child. The interviewee recorded notes after each interview. Interviews with trial participants ranged from 4.5 to 21.5 min, and those with staff were 19.5 to 57.6 min.

The trial participant interview schedule covered the participants' experience of recruitment and participation in the trial. The staff interview schedule included the staff's experience of recruitment and their overall experience of delivering the trial at their site.

All interviews were undertaken by telephone. Data saturation was discussed between RC and AP separately for staff and trial participant interviews.

### Data analysis
An inductive thematic analysis was used to analyse the data, which was undertaken within a phenomenological framework. We used Braun and Clarke's reflexive thematic analysis framework, which involves six phases of analysis—familiarisation, initial coding, theme construction, reviewing themes, defining themes and producing the report.[17] Two researchers (AP and RC for the trial participant interviews, and PK and RC for the staff interviews) developed the coding structure separately by coding a small number of transcripts (four for the participant interviews and two for the staff interviews). NVivo (V.12) was used to undertake all analyses. The researchers then met and resolved any differences in coding in order to develop the final list of codes. All other transcripts were then coded by one researcher, either AP or RC for the trial participant interviews, and RC or PK for the staff interviews. The coding trees were formed of overarching themes that were related to the overall aims of this qualitative study—the experience of receiving/delivering ES, reasons for participants deciding not to receive ES after being randomised to do so, preconceptions of the ES procedure, and experiences of recruitment. However, in this manuscript, the experience of receiving/delivering ES is not reported in order to reflect only those themes that have the potential to improve the experience of participants in future studies. RC undertook all analysis steps after coding, including formation of themes and writing of the report. Interviewees did not feedback on the findings or transcripts. After analysing the transcriptions no new themes emerged, thus, data saturation was determined to have been reached.

### Patient and public involvement
Input into the design of the study, interview schedules and patient facing materials were sought from the Sheffield Reproductive Health Research Public Advisory Panel and the trial's public involvement representative (AS).

## RESULTS
### Interviews and characteristics of participants
Forty-four trial participants were approached to participate in the study, of which two actively declined to participate and 15 either did not respond or were not contactable. As a result, 27 trial participants took part.

Interviewed participants had an average duration of infertility of three years (range 11 months to 8 years 6 months) and were interviewed, on average, 17.8 months after they were randomised into the trial (range from eight to 33 months). Most participants were in the 33–37 years age range, with ages ranging from 21 to 37 years. Participants were recruited from National Health Service (NHS) sites, except for four participants, who were recruited from a privately run centre (site six). The demographics of the trial participants that took part in this study are summarised in table 2.

Seven site staff (one principal investigator (PI) and six research nurses) took part. Fifteen participants were invited (seven PIs and eight research nurses)—one participant actively declined due to lack of time, five participants did not reply following multiple contact attempts. Interviews were attempted to be scheduled for two individuals but a mutually agreed time could not be found. The characteristics of the participating site staff can be found in table 3.

Trial participants and site staff discussed four major themes during the interviews—the experience of recruitment to the trial, the written materials that were provided at recruitment, the lack of equipoise in both trial participants and site staff and the demoralising effect this had on some participants randomised to the control arm, and the reasons for withdrawing from the ES once randomised to receive it. Below, we discuss these themes.

**Table 2** Characteristics of Interviewed trial participants

| ID | Age range at randomisation | Live birth recorded? | Randomisation arm |
|---|---|---|---|
| Site 1 | | | |
| Participant 1 | 33–37 | No | ES |
| Participant 2 | 33–37 | Yes | ES |
| Participant 3 | 27–32 | No | ES |
| Participant 4 | 27–32 | No | ES |
| Participant 5 | 27–32 | No | ES |
| Site 2 | | | |
| Participant 1 | 27–32 | Yes | ES |
| Participant 2 | 27–32 | Yes | ES |
| Site 3 | | | |
| Participant 1 | 33–37 | No | ES |
| Site 4 | | | |
| Participant 1 | 33–37 | No | ES |
| Participant 2 | 33–37 | Yes | ES |
| Participant 3 | 33–37 | Yes | TAU |
| Participant 4 | 33–37 | No | ES |
| Site 5 | | | |
| Participant 1 | 27–32 | Yes | ES |
| Participant 2 | 27–32 | Yes | TAU |
| Participant 3 | 21–26 | No | ES |
| Site 6 | | | |
| Participant 1 | 27–32 | Yes | TAU |
| Participant 2 | 27–32 | Yes | ES |
| Participant 3 | 33–37 | Yes | ES |
| Participant 4 | 27–32 | No | TAU |
| Site 7 | | | |
| Participant 1 | 33–37 | No | ES |
| Participant 2 | 33–37 | Yes | ES |
| Participant 3 | 21–26 | No | ES |
| Participant 4 | 33–37 | Yes | TAU |
| Site 8 | | | |
| Participant 1 | 27–32 | No | TAU |
| Participant 2 | 33–37 | Yes | ES |
| Participant 3 | 27–32 | No | ES |
| Participant 4 | 33–37 | No | TAU |

ES, endometrial scratch; ID, identification number; TAU, treatment as usual.

### Experience of the general recruitment process

Trial participants were generally happy with the recruitment process.

*I think it was all done really well. It was just, it was really well explained and I don't … think, if it was happening again, … I wouldn't change anything about it.* Participant 2—Site 8

Participating in research while undergoing IVF was seen by many as 'exciting' and 'interesting', and provided an alternative focus for two participants.

*I think it's in a way it was actually it was quite a good thing because I think it gave me something else to focus on […] it was quite nice to have something that was being discussed where I felt actually this has got a wider benefit.* Participant 4—Site 4

Three participants felt that the approach to take part in the trial was too informal, with the participants either having to ask about the trial, or the trial being mentioned at the last minute when there were previous opportunities to raise this.

**Table 3** Characteristics of interviewed site staff

| ID | NHS/private | Gender |
|---|---|---|
| PI-SITE1 | NHS | Female |
| RN-SITE1 | | Female |
| RN-SITE3 | NHS | Female |
| RN-SITE4 | NHS | Female |
| RN-SITE5 | NHS | Female |
| RN-SITE6 | Private | Female |
| RN-SITE9 | NHS | Female |

NHS, National Health Service; PI, principal investigator; RN, research nurse.

*I don't know if it should be more formal really, because it was just kind of, if they remembered on that day that I would be a possible candidate …, maybe you should get it in a pack or something at the beginning.* Participant 2—Site 1

### Written information provided at recruitment

Four participants could not recall the recruitment materials, due to the time that had elapsed since they were recruited. Of those that could, five participants stated that the materials were clear and informative.

*I thought the information sheet was really useful and informative.* Participant 4—Site 7

There were differences in opinion with regards to the amount of information that trial participants were asked to read during the recruitment process, with some participants stating that the materials were not too onerous to read.

*I think it was maybe like a double sided page it wasn't kind of too much.* Participant 2—Site 2

However, others reflected on the amount of information provided more negatively, with two participants discussing the amount of information they received as 'overwhelming'.

*It can be very overwhelming, having lots of trial information given to you at the same time as all your IVF treatment. It just means that your pack of information to read through, I think I had about twenty different documents or leaflets.* Participant 4—Site 7

One participant suggested that the PIS should contain more information about related studies.

*After reading about other studies in the newspapers like The Guardian, it might have been also interesting to hear a little bit about other studies that were going on… Because, I think when I did read about this study in The Guardian…and it suggested that it wasn't helpful. There was part of me that I think became a little bit angry that maybe I'd put myself through it at the time.* Participant 2—Site 7

### Equipoise

Participant's lack of equipoise were exemplified by eighteen of the interviewed participants stating that a major reason to participate in the trial was to increase their chances of a positive outcome from their IVF cycle.

*Obviously if they'd have said to us the Scratch trial probably won't have improved our chances of getting pregnant, I probably wouldn't have taken part. So we did it on the basis that we wanted to be on the clinical sample so that we could improve our chances of getting pregnant.* Participant 1—Site 7

Outside of the trial, the privately run site (site 6) offered the ES for a fee, whereas the NHS sites did not offer it at all to women undergoing their first IVF cycle. Two of the four participants from the private site participated in the trial to receive a 'free' ES. However both of these participants seemed to be well informed regarding the lack of evidence to support the use of ES in this population.

*They went through all the extras that could be added on to our treatment…they talked about the fact that they were involved in the scratch trial and that it was our first round of IVF. We thought that was probably a good way to do it, rather than to pay for it, if we get lucky and we have it then brilliant and if we don't, you don't yet know how successful it is.* Participant 3—Site 6

There seemed to be a lack of understanding of the potential for the ES to negatively affect the chances of live birth, with five trial participants believing that the ES procedure could not 'harm' them.

*There wasn't anything that could potentially harm my chances of success. It wasn't, you know, it wasn't like trying out a new medicine or anything like that.* Participant 2—Site 8

### Source of positive preconceptions

Trial participants discussed that a small number of site staff supported their positive preconceptions, sometimes in a subtle way.

*You don't know whether it helps or not but there seems to be this influence from some within the clinic that almost nudge nudge, wink wink, you know,' you're gonna be fine because this really does help.* Participant 2—Site 7

In other cases, the trial participant reported that they believed site staff thought ES would increase their changes of a live birth

*Basically they said: 'It's completely up to you, it's something that we're giving to you, you can decide.' … obviously they do believe scratches increase chances of success.* Participant 3—Site 7

However, for some participants, the information provided by site staff ensured they were aware that there

was insufficient evidence to suggest that the procedure will improve their chances of success.

*The consultant suggested that there wasn't much evidence about whether the scratch was helpful or not when it comes to success of IVF.* Participant 1—Site 8

Participants also sought information from other sources, most frequently from the internet. For five participants, the internet provided the participant with positive preconceptions of the effect of the ES on their upcoming IVF cycle.

*Just the success rate for the people who were trying to do IVF; who were trying to have babies. The success rate after the scratch was high.* Participant 2—Site 5

However, for two participants, reading information on the internet led them to a position of equipoise.

*I wasn't sure, because some studies said it was successful, others said it wasn't, so I was on the fence.* Participant 2—Site 1

### Discontent when randomised to the control arm of the trial

Positive preconceptions of the effect of the ES on the outcome of their IVF cycle led five out of the eight interviewed participants randomised to the control arm feeling some level of discontent with the outcome of randomisation. Some participants discussed feeling slight disappointment.

*I think there was a slight bit of disappointment which is ridiculous really because it's randomised. I wasn't sort of like devastated or anything like that… It was, that's what current practice is, to not have the scratch for a first time if under 35 so I was just like well we'll go with that and see what happens. So I didn't dwell on it or anything like that.* Participant 4—Site 7

For others, randomisation to the control group triggered significant demoralisation.

*I was really quite disappointed and quite emotional when I, didn't get on the Scratch trial itself because I was worried that my chances of getting pregnant were minimized.* Participant 1—Site 7

### Staff awareness of positive preconceptions and demoralisation

Site staff were aware of the trial participant's positive preconceptions.

*They would generally say 'oh I've heard about the procedure and it can benefit the chances of you getting pregnant'.* RN-SITE4

In order to attempt to dispel any positive preconceptions during the recruitment process, site staff spent time explaining the lack of an evidence base for ES, and the need for an RCT, to potential trial participants.

*We did explain to them comprehensively the evidence, around the Endometrial Scratch and the requirement for a big trial like this to be conducted for us to be able to know the answers and to know whether they should be offered to all patients going through forward for IVF.* PI-SITE1

However, site staff reported that participants seemed to struggle to understand the need for a control arm when some studies reported a benefit to having the ES.

*Sometimes patients found it difficult to accept that if there has been some study that indicate potential benefit, then patients found it difficult to accept that they could be randomised into the control arm.* PI-SITE1

Many site staff were also aware of some participant's disappointment of being randomised to the control arm.

*Some people did express great disappointment and in particularly that way round actually it was nearly always the ones that had wanted it.* RN-SITE6

Staff described mechanisms they had developed for dealing with this disappointment, summarised in table 4.

### Reasons for withdrawing from the ES

Seven participants were interviewed that, having been randomly allocated to receive the ES, did not receive it. None of the reasons for withdrawing were related to equipoise; five participants declined the ES due to reasons that were out of the control of their fertility centre (illness, holidays and work commitments). Two participants stated reasons for declining ES that could have been prevented. For one, the site forgot to check the participant's records for a recent smear test result.

*We were scheduled to start the IVF and then I got a call from one of the nurses and she said 'Oh I've just checked your records and it turns out you're overdue a smear test, we can't proceed with the IVF until you've had a smear test.* Participant 1—Site 5

For the other, the site did not tell the participant to refrain from sexual intercourse in the menstrual cycle in which the ES was being performed. This participant recommended that discussions around abstinence should be more prominent in the recruitment process.

*I think that, that discussion [abstinence from sexual intercourse] needs to be kind of one of the leading things.* Participant 1—Site 1

## DISCUSSION AND IMPLICATIONS
### Main findings

This qualitative study demonstrates that participants recruited to the ES Trial were, on the whole, happy with the trial recruitment practices. A small number of participants felt that the invitation to participate in the trial was too informal, and a minority described the amount of written material they received as 'overwhelming'. There appeared to be a lack of equipoise among both sets of

**Table 4** Methods site staff used to deal with participant's disappointment at being randomised to the control arm of the trial

| Method of dealing with disappointment | Quote |
|---|---|
| Explain the importance of the control group | 'So I think its explaining the importance of being in that control group to them. And how we can, you know, we don't know if there was any benefit until we've actually, that's the whole purpose of doing the trial, to answer a question.' RN-SITE1 |
| Describe the lack of evidence to support ES | 'Try and sort of explain that, there was no evidence whether it was a benefit anyway, and that normally if they were going through their treatment, I understand they wouldn't be having a Scratch, so it wasn't as if they missing out on anything.' RN-SITE6 |
| Explain the increased contact with staff that participating in research allows | 'If it's the control you know it's a bit of extra contact with people and I would always say 'look I'm here in the unit, you're not having the Scratch, but if I can help you in any other way, feel free to ring the research mobile'. You know to try and always make it I suppose to soften the blow really, and for those who were disappointed.' RN-SITE9 |

ES, endometrial scratch.

interviewees with regards to the potential effect of the ES on the outcome of the upcoming IVF cycle. Trial participants had positive preconceptions of the ES procedure, which often originated from their own internet research, and in some instances, participants indicated that they believed site staff supported these preconceptions. Some trial participants appeared to not be aware that receiving the ES could potentially reduce their chances of a successful IVF outcome. This, coupled with the positive preconceptions of the ES, led to dissatisfaction among some participants randomised to the control arm. Participants from the private centre had a similar experience to those recruited from NHS centres, except that those recruited from the private centre stated a reason to participate in the trial was to receive a free ES, which at the time, was offered at an extra cost by their centre.

### Strengths and limitations

A strength of this study is that it included a purposeful sample of women that participated in a large RCT, ensuring variation in trial arm (control or intervention), and live birth status. We also interviewed staff in order to seek their experiences. A limitation includes that only those that participated in the trial were interviewed, with those participants that withdrew from the trial not being interviewed. This may result in a skewed reflection of the experience of recruitment to the trial. In addition, participants were interviewed, on average, 17.8 months after they agreed to participate in the trial, and furthermore, were interviewed after their IVF cycle had completed. This therefore results in participants reflections on their experiences being clouded by time, and being affected by the outcome of their IVF cycle.

### Comparison with existing literature

Our study mirrors the findings of other studies that have described that potential participants have different information needs, and may require different versions of the PIS;[5 6] the discontent felt by participants when randomised to the control group of an RCT;[18 19] participants seeking

information from internet sources;[14] and the motivation to receive the intervention has been reported as a major reason participants decide to take part in interventional trials.[20 21] In this study, site staff tried to reposition the participant's positive preconceptions of the ES by explaining the evidence base, which has also been reported in other studies.[15 20] However, it appears that in some cases, participants received cues from site staff that the ES may benefit them, despite having no evidence to support this. Staff have been found to communicate equipoise poorly in other studies, including providing inbalanced descriptions of trial treatments and predicting RCT outcomes.[16]

Other studies have also reported that participants overestimate the benefits, and underestimate the risks, of participating in research.[1 2] Our study adds to this evidence base by reporting that, in our study, participants 'arrived' with positive preconceptions, which, in the most part, originated from the internet. This, coupled with subtle signals from recruiting site staff, led to positive preconceptions of the potential effect of the ES procedure on their IVF outcome, and potentially an underestimation of the potential harms of the procedure. This may have been exacerbated by the time-sensitive nature of fertility treatment and the desire for a positive outcome; both clinicians and patients have been reported to feel the need to improve the outcomes of treatment immediately following a failed treatment cycle.[22]

### Implications

Recruiting staff should be aware that participants may base their decision to participate in an RCT on internet research, which risks participants making their own subjective interpretation of the evidence, and may result in overly positive preconceptions of the potential benefits—and a downplaying of the potential harms—of receiving the novel intervention.

Staff should also be aware of subtle cues that may suggest to the participant that the trial intervention may

be beneficial to them. The control group should be introduced with similar emphasis to the intervention group, however, discontent once randomised to the control group may still be felt—we present three ways that this was lessened by site staff in our trial (see table 4). Other studies have provided recommendations to guide 'recruitment conversations' in surgical trials—in the opinion of the authors, these guidelines are applicable to studies outside this specialty.[23–25]

In order to improve the information provided to participants during the informed consent process, some participants may require more detailed information about the research literature, including the potential harms of the intervention being tested. The authors reflect that the ES trial PIS did not explicitly state the evidence for or against the ES procedure, nor did it explicitly state the ES may potentially harm the chances of a successful IVF outcome. PPI input was sought into the PIS, but PPI representatives did not comment on the lack of a discussion of the potential harms.

Improvement to trial participant's understanding of the harms of taking part in research, and the role of the control group, may improve the quality of informed consent. This, in turn, may lead to reduced attrition from the trial, especially differential drop-out, where a different proportion of participants drop-out from one trial arm.[26] Reduced attrition may lead to improved external validity of the trial.[27]

## Future research

Future research should focus on how the potential harms of taking part in a study can be relayed to participants, in order to deal with overly positive preconceptions. Details of the potential harms of participating in a study have to be clearly stated in the PIS in clinical trials of an investigational medicinal product (CTIMPs). This issue may have not been given enough consideration in non-CTIMPs; such information could be reviewed by research ethic committees prior to providing ethical approval. Checklists to assist PPI representatives in reviewing important patient facing documentation (including a prompt to ensure the potential harms are adequately discussed) could be devised.

Other methods of providing information about the trial to potential participants should be explored, possibly by providing participants with a choice of levels of information, as suggested in previous studies.[6 28] The use of multimedia to inform participants has been assessed within various studies with a trial, finding that such mediums did not significantly impact on recruitment to the trial.[29 30] However, the 'quality' of informed consent, and the participant's understanding of the research, was not assessed. Patient acceptable alternatives to the patient information sheet should be investigated, in order to reduce the burden on patients at the start of their fertility treatment.

**Acknowledgements** We would like to thank those trial participants and site staff that gave their time and participated in the qualitative interviews. We would also like to thank the Reproductive Health Research Public Advisory Panel at the Jessop Wing, Sheffield Teaching Hospital NHS Foundation Trust, for their input into the design of this qualitative study.

**Contributors** RC designed the study and led on data collection, analysis and the first draft of the manuscript. DW and CP provided input into the design of the study, analysis and reviewed and approved the manuscript. AS provided input from a lay-person perspective into the design of the study, patient facing materials, results and manuscript. AP inputted into the design of the study, assisted with data collection and analysis, and reviewed and approved the final version of the manuscript. PK assisted with data analysis, and reviewed and approved the final version of the manuscript. MM inputted into the design of the study and reviewed and approved the final version of the manuscript.

**Funding** This study is funded by the National Institute for Health Research (NIHR) (Health Technology Assessment Programme (project reference 14/08/45)).

**Disclaimer** The views expressed are those of the author(s) and not necessarily those of the NIHR or the Department of Health and AQ5 Social Care.

**Competing interests** None declared.

**Patient consent for publication** Not required.

**Ethics approval** This qualitative substudy was approved by Berkshire South Central Research Ethics Committee (REC) (16/SC/0151). All participants gave informed consent before taking part.

**Provenance and peer review** Not commissioned; externally peer reviewed.

**Data availability statement** No data are available. No data are available. The experiences expressed during the interviews refer to sensitive and personally relevant stories; it was agreed that transcripts would not be shared with other researchers in order to preserve the anonymity of study participants.

**ORCID iD**
Robin Chatters http://orcid.org/0000-0002-1945-6011

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
