## [Reviewer comments · BMJ Open]

ARTICLE DETAILS

TITLE (PROVISIONAL)	The experiences of trial participants and site staff of participating in and running a large randomised trial within fertility (the Endometrial Scratch Trial) – a qualitative interview study
AUTHORS	Chatters, Robin; White, David; Pye, Clare; Petrovic, Ana; Sizer, Anya; Kumar, Pavithra; Metwally, Mostafa

VERSION 1 – REVIEW

REVIEWER	Darbyshire, Julie University of Oxford, NDCN
REVIEW RETURNED	20-May-2021

GENERAL COMMENTS	This was a nicely written, clear account of a post-study qualitative evaluation of patient and staff experiences of participating in/delivering an IVF RCT. I'm not entirely convinced by the statement that 27 is a "large" number but would say that this is a qualitative study and numbers are not important so this isn't a necessary statement to make in this context. The key recruitment point is purposeful sampling and data saturation. The authors make good use of participant quotes in the results section and have made intelligent and sensible decisions about how to present the themes identified across the two groups of participants. Lessons learned from the recruitment and participation experiences are valid for all RCTs, where it is common for participants to believe that the test intervention is better than usual care. I would encourage the authors to explore these wider implications of their results more than they do. In the data analysis section the authors explain that they are only including themes identified from the interviews that are relevant for changing future practice. I strongly support the inclusion of themes associated with reasons for declining the trial intervention when participants were randomised for ES. Insight into participant acceptance of trial treatment would be incredibly valuable for all kinds of intervention. As PK and AS were also involved in the analysis, please add them to 'research team/reflexivity' section. From table 4 I learned that not all study sites were NHS. It feels important that the authors make some comment on the similarity
---

	(or otherwise) of themes, preconceptions, or experiences raised by patient participants from site 6 (and other non-NHS sites, if any) compared with those from NHS sites. In the final section the authors report that the trial participant information sheet wasn't adequate in a few ways. The evidence for this needs to be available to the reader; ideally as text in the results section but could be referenced from the results section as supplementary material if word count is an issue. Some comment on the input from the public involvement representative at the PIS draft stage would also be useful here to guide patient materials development in future trials. In addition to offering potential participants a choice of information level, there is also a need for guidance in recruitment materials in different formats and the authors could bring this into their implications section. Adding the REC approval ref/ISRCTN registration would be easy to add and saves the reader looking it up/hunting down the reference for the protocol publication. There are two typos (sorry): page 8, line 58 (changes should read chances) and page 12, line 15 (provide should read provided) Table 3 has a lot of information. The participant ID could be presented in a simpler way by listing site first and then participant numbers within each individual site 'block', eg: Site 1 Participant 1 Participant 2 Site 2 Participant 1 Participant 2 etc Similarly the 'role' column in table 4 is unnecessary as this is obvious from the ID.
--	---

REVIEWER	Gibreel, Ahmed
REVIEW RETURNED	Mansoura University, Obstetrics and Gynecology Department 22-May-2021

GENERAL COMMENTS	This is a well written paper. One of the addressed limitation is the long time between randomisation and interviewing.
--

VERSION 1 – AUTHOR RESPONSE

REVIEWER: 1

Comment: I'm not entirely convinced by the statement that 27 is a "large" number but would say that this is a qualitative study and numbers are not important so this isn't a necessary statement to make in this context. The key recruitment point is purposeful sampling and data saturation.

Response: We agree with the reviewer and have removed the statement from page 12 that suggests the study was large.

Comment: Lessons learned from the recruitment and participation experiences are valid for all RCTs, where it is common for participants to believe that the test intervention is better than usual care. I would encourage the authors to explore these wider implications of their results more than they do.

Response: We have added to the implications section of the discussion, broadening the implications to RCTs in general.

Comment: In the data analysis section the authors explain that they are only including themes identified from the interviews that are relevant for changing future practice. I strongly support the inclusion of themes associated with reasons for declining the trial intervention when participants were randomised for ES. Insight into participant acceptance of trial treatment would be incredibly valuable for all kinds of intervention.

Response: We have added the reasons for discontinuation from the intervention to page 11 of the manuscript.

Comment: As PK and AS were also involved in the analysis, please add them to 'research team/reflexivity' section.

Response: This has been added to page 4.

Comment: From table 4 I learned that not all study sites were NHS. It feels important that the authors make some comment on the similarity (or otherwise) of themes, preconceptions, or experiences raised by patient participants from site 6 (and other non-NHS sites, if any) compared with those from NHS sites.

Response: The main difference between experiences reported by those participant recruited at private centres, and those at NHS centres, was that those at private centres stated they participated in the trial in order to receive a free endometrial scratch (ES), which the private centre was charging for at the time. We have added this to page 8 and 9 of the results.

Comment: In the final section the authors report that the trial participant information sheet wasn't adequate in a few ways. The evidence for this needs to be available to the reader; ideally as text in the results section but could be referenced from the results section as supplementary material if word count is an issue.

Response: The comment on page 13 of the manuscript that states the inadequacies of the PIS is a reflection by the authors that the participants were not aware of the harms of the research, and this may have been impacted by the fact the PIS did not clearly state the harms. This section, on page 13, has been altered in order to clarify that this is in fact the opinion of the authors, and has not been gained from the qualitative data.

Comment: Some comment on the input from the public involvement representative at the PIS draft stage would also be useful here to guide patient materials development in future trials.

Response: We have added this to page 13 of the manuscript. Essentially, PPI representatives reviewed the PIS, but did not pick on the fact that it did not adequately discuss the potential harms of the research. Therefore, in the discussion, we have suggested that a checklist could be put together to assist PPI representatives in reviewing key patient facing documentation.

Comment: In addition to offering potential participants a choice of information level, there is also a need for guidance in recruitment materials in different formats and the authors could bring this into their implications section.

Response: We have added this to the implications section, page 14.

Comment: Adding the REC approval ref/ISRCTN registration would be easy to add and saves the reader looking it up/hunting down the reference for the protocol publication.

Response: We have added the REC reference to the ethics section on page 5. The ISRCTN registration is stated within the abstract.

Comment: There are two typos (sorry): page 8, line 58 (changes should read chances) and page 12, line 15 (provide should read provided).

Response: Thank you for identifying these typos – they have been corrected.

Comment: Table 3 has a lot of information. The participant ID could be presented in a simpler way by listing site first and then participant numbers within each individual site 'block', eg:

Site 1

Participant 1

Participant 2

Site 2

Participant 1

Participant 2

etc

Response: We have altered the table as per the reviewer's comments.

Comment: Similarly the 'role' column in table 4 is unnecessary as this is obvious from the ID.

Response: We agree and have removed the 'role' column.

REVIEWER: 2

Comment: This is a well written paper. One of the addressed limitation is the long time between randomisation and interviewing.

Response: We agree that this is a limitation of this work and have stated this as a limitation within the "strengths and limitations of this study" and discussion sections.

VERSION 2 – REVIEW

REVIEWER	Darbyshire, Julie University of Oxford, NDCN
REVIEW RETURNED	29-Jul-2021

GENERAL COMMENTS	The authors have addressed the queries raised by the reviewers. Thank you for responding to each comment individually and making it clear what you changed in the manuscript, and where. I have no further queries. This article has value and lessons for all clinical research where patients are recruited, beyond the immediate field of fertility treatments. I recommend this manuscript for publication.
---